# Immune Microenvironment Landscape in CNS Tumors and Role in Responses to Immunotherapy

**DOI:** 10.3390/cells10082032

**Published:** 2021-08-09

**Authors:** Hinda Najem, Mustafa Khasraw, Amy B. Heimberger

**Affiliations:** 1Department of Neurological Surgery and Northwestern Medicine Malnati Brain Tumor Institute of the Lurie Comprehensive Cancer Center, Feinberg School of Medicine, Northwestern University, Chicago, IL 60611, USA; hinda.najem@northwestern.edu; 2The Preston Robert Tisch Brain Tumor Center, Duke University, Durham, NC 27710, USA; mustafa.khasraw@duke.edu

**Keywords:** glioma, CNS metastasis, immune composition, tumor microenvironment, immune therapy, immune checkpoints, T cells, tumor associated macrophages/microglia

## Abstract

Despite the important evolution of immunotherapeutic agents, brain tumors remain, in general, refractory to immune therapeutics. Recent discoveries have revealed that the glioma microenvironment includes a wide variety of immune cells in various states that play an important role in the process of tumorigenesis. Anti-tumor immune activity may be occurring or induced in immunogenic hot spots or at the invasive edge of central nervous system (CNS) tumors. Understanding the complex heterogeneity of the immune microenvironment in gliomas will likely be the key to unlocking the full potential of immunotherapeutic strategies. An essential consideration will be the induction of immunological effector responses in the setting of the numerous aspects of immunosuppression and evasion. As such, immune therapeutic combinations are a fundamental objective for clinical studies in gliomas. Through immune profiling conducted on immune competent murine models of glioma and ex vivo human glioma tissue, we will discuss how the frequency, distribution of immune cells within the microenvironment, and immune modulatory processes, may be therapeutically modulated to lead to clinical benefits.

## 1. Introduction

Gliomas are the most common primary brain tumors and are classified by the World Health Organization (WHO) as grade I–IV tumors based on molecular and genomic features, allowing a more accurate classification of patients. Given the transition from histological characterization, this newer molecular classification system enables precision medicine therapeutic selection and leads to a more accurate prediction of prognosis [1,2]. One of the most central genetic characteristics is the isocitrate dehydrogenase mutation (IDH1) status that is commonly expressed in low-grade gliomas and reflects a favorable prognosis relative to IDH1 wild-type gliomas that are high-grade glioblastomas [3,4]. Glioblastoma is particularity challenging to treat despite multi-modal therapy, and median survival is 14.6 months [4,5].

Despite the beneficial effects of immunotherapies in multiple types of cancers including brain metastases from several solid tumors [6,7], the vast majority of glioma patients do not benefit. In fact, there is a lack in the understanding of glioma microenvironment that needs to be further elaborated in order to explain the hindering effects in the use of the available immunotherapies in the treatment of glioblastoma. Recent studies have shown that the immune composition and states are unique and specific to cancer lineage [8,9]. Additionally, there are differences in immune suppressive pathways and immune targets not only between lineages [10], but also between individual patients harboring the same types of cancers [11]. Further abrogating potential immune effector responses in gliomas, are a wide variety of immune suppressive secreted factors, exhaustion and sequestration of T cells in the bone marrow, administration of steroids, and the blood–brain barrier (BBB) that have been recently reviewed [12].

## 2. Why Current Immunotherapies Have Shown Insufficient Outcomes in Gliomas

Several criteria need to be met in order to induce effective clearance of cancer, including: the presence of the immunological target, capacity to induce immune activation, homing and dispersal of effector immune cells throughout the tumor microenvironment (TME), and the capacity to maintain an effector response. In the context of glioblastoma, multiple barriers exist for each criterion that are further confounded by an overwhelming recruitment of immunosuppressive cells, inhibition and exclusion of effector anti-tumor cells, secretion of immunosuppressive factors (like TGF-β) and expression of inhibitory checkpoint markers [13].

Gliomas, especially glioblastoma, have not been responsive in general to immune checkpoint inhibitors and one reason is that the biomarkers of response such as PD-L1 expression, tumor infiltrating lymphocytes (TILs), and neoantigen creating defects, are relatively lower compared to other responsive cancers [14,15,16]. The low frequency of TILs [10,17] is likely due to their sequestration in the bone marrow [18], exhaustion, and their refractory nature to being reinvigorated with immune checkpoint inhibitors [19] and other immune modulators [11]. In the setting of brain metastases, T cell priming to tumor antigens is already effectuated systemically even before the appearance of brain metastasis [20]. This can explain the pool of TILs in brain metastasis and the effective immune response created against the tumor following successful immune checkpoint inhibitor immunotherapy. Most immune targeted therapies have been focused on modulating the T cell as they represent the main effectors of tumor cell cytotoxicity in the vast majority of cancers. However, they constitute a small portion of the immune population in the glioma microenvironment. The immune microenvironment of glioma is predominately infiltrated with myeloid-derived cells, like macrophages/microglia, that express immunosuppressive phenotypes, and have been shown to enhance tumor progression and are associated to poor survival rates [10,21]. The heterogeneity of the immune cells and the heterogeneity of their immunosuppressive phenotypic expressions makes it difficult to generate and maintain an effective immune response, and it represents a major cause of resistance to immune checkpoint inhibitors.

Although chimeric antigen receptor (CAR) T cell immunotherapy would be able to overcome several of the aforementioned limitations, in contrast to the clonal expression of tumor antigens on leukemia that provides a more uniform and durable response, no ubiquitous homogenously expressed antigen in GBM has been identified to date and as such CAR therapy targeting a tumor-specific antigen, epidermal growth factor variant III (EGFRvIII), did not result in durable responses and antigen escape resulted [22] similar to prior clinical trials of EGFRvIII specific peptide vaccines [23].

## 3. The Unique Immune Microenvironment of Gliomas

Gliomas, especially glioblastomas, are typically described as having a “cold” microenvironment given the paucity of TILs displaying an effector response. Molecularly distinct gliomas, depending on their IDH mutation status, have different immune compositions and landscape that defines its TME [24,25,26]. IDH-mutant gliomas are almost totally devoid of TILs in comparison with brain metastasis that are highly enriched with activated and exhausted T cells [8,27]. This status of low infiltration of T cells in gliomas, especially in the IDH-mutant subtype, creates an environment with low expression of immune checkpoint targets, which provides one possible reason for the resistance to immune checkpoint inhibitors. On the other hand, IDH-mutant tumors are enriched with tumor-resident microglia, which is in contrast to IDH wild-type and brain metastasis that are infiltrated with monocyte-derived macrophages originating from the periphery [8,26,27]. In contrast to brain metastasis, 40% to 70% of the total immune cell populations in glioblastoma are myeloid derived cells, representing the most abundant immune cell type infiltrating these tumors [28] that typically have an immune suppressive phenotype.

In addition to the different immune populations that populate different cancer lineages within the CNS, unbiased transcriptional profiling is beginning to reveal distinct subpopulations and states that further increase immunological complexity and heterogeneity even within the same immune cell lineage in the TME. Based on transcriptomics, there may be many different microglia subtypes that demonstrate different functional signatures such as phagocytosis, antigen presentation and lytic functions [29]. Although conventional described in the context of M1 (tumor suppressive/immune supportive) or M2 (tumor supportive/immune suppressive) phenotypes, ex vivo data of gliomas have revealed that neither of these were adequate descriptions for tumor associated macrophages (TAMs) [30]. With emerging single cell sequencing data, distinct subtypes/subgroups will be emerging that were not evident with bulk profiling initiatives. The innate immune cells in gliomas likely express vastly different molecular phenotypes, transcriptional states and functionalities that are influenced by cancer lineage (gliomas or metastasis), genetic status (IDH-mutation status), and within the TME location (hypoxia) further confounding the heterogeneity of the TME [30,31] likely even within the same tumor. This area needs further investigation, including functional immune characterization, in order to comprehend the different subtypes of the various tumor associated immune cells and to be fully informed regarding therapeutic selection and optimization.

Additional evidence that the glioma TME is distinct from other cancer lineages is the lack of tumor mutational burden (TMB) being a predictive response biomarker to immune checkpoint inhibitors. McGrail et al. recently showed in an analysis of over 10,000 patients that not all tumors with high-TMB developed effector immune responses to immune checkpoint inhibitors therapy [32]. Specifically, in breast, prostate, and glioma cancer patients there appears to be no correlation between CD8 T cell levels and neoantigen load, with overall survival [33]. In another analysis of recurrent glioblastoma patients, a very low-TMB state was associated with better response to immune checkpoint inhibition or polio virotherapy [34]. The distinction may reside in the mutational composition found in glioma not being particularly immunogenic or alternatively that the mutagenic antigens are eliminated potentially due to better immune surveillance in the patients with lower TMB, resulting in better immune recognition and elimination of mutated subclones before the initiation of immunotherapy. Such immune environment would keep the overall TMB low and would explain the otherwise apparent discordance between studies of patients with constitutional DNA mismatch repair syndromes that show responses to immunotherapy. Other investigators have shown that specific mutations like PTEN, are associated with worse outcomes to immune checkpoint inhibitors and one could consider removing these subjects from use of immune checkpoint inhibitors [35], rather than enrichment selection based on TMB. In contrast, in recurrent glioblastoma patients that respond to PD-1 blockade, there is enriched expression of BRAF and PTPN11 activation mutations [35,36]. Arietta et al. showed that phosphorylated ERK1/2 (p-ERK1/2) proteins are predictors of response in recurrent glioblastoma patients to PD-1 inhibition even in absence of BRAF and PTPN11 expression [37]. Given the complexity of the steps needed to mount an effective anti-tumor immune response, it is likely that prediction of responsiveness to immunotherapies will not be predicated on a single marker. A systems biology approach possibly using different analysis platforms, through genome sequencing, emerging current single cell analysis, immune landscape observation and functional analysis may need to be consolidated to ultimately predict responses and ultimately understand this unique microenvironment.

## 4. Immune Landscape Distribution

The heterogeneity of the immune microenvironment of gliomas is not only defined by a different immune compositions within a tumor, but also the localization throughout the tumor and its surrounding. Most studies focused on the immune profiling within the tumor itself, neglecting its surrounding. A chemokine analysis from the Ivy Atlas suggests differential expression of immune populations at the infiltrating edge [38]. The CNS borders are populated by myeloid cells from adjacent bone-marrow niches that supply innate immune cells under homeostatic and pathological conditions [39]. Additionally, there appears to be a predilection of T cells in the perivascular tumor regions [40]. This area of investigation could provide additional insights into the mechanisms of failure of immunotherapy and strategies that could be employed to overcome them.

Current dogma is that antigen presentation by antigen presenting cells (APCs) for T cell activation occurs in the lymph node with the effector response migrating to the TME. Under certain treatment scenarios, such as radiation and signal transducer and activator of transcription 3 (STAT3) blockade, dendritic cells are drawn into the TME forming immune activating immune cluster interactions that ultimately result in tumor clearance from the CNS [41]. Additionally, when APCs are deposited into the TME with BBB opening ultrasound genetically modified with T cell chemokines, a therapeutic result is seen in preclinical models [42]. Pilot data is beginning to emerge that immunological interactions within the TME are different between cancer lineages. For example, CD3 T cells interact with many different types of other immune cells and in clusters in brain metastasis whereas in gliomas, they are usually solitary [43]. If these cluster interactions are immune activating events, then the T cells may not yet be exhausted and therefore still susceptible to immune modulation with immune checkpoints. It is likely that the immune interactome varies throughout the different TME compartments of the tumor and the TME niches from the necrotic core, tumor, to the CNS interface. Although the focus has been on interrogation of the tumor, en bloc surgical resections could be exploited to study the content, spatial distribution, and interactions of immune cells using multiplex immunohistochemistry and/or mass cytometry. The evolution in the way we visualize the immune interactome in the TME may ultimately reveal patient specific interactions. Whether immune reactivity in the TME could be interrogated using advanced brain tumor imaging with either magnetic resonance imaging (MRI) or PET [44,45] remains an area of future investigation which is much needed for the further evolution of the field given the risks and limitations of longitudinal biopsy/resection.

## 5. The Role of the BBB in Immunotherapies

The BBB represents an important factor of treatment failure at the infiltrating edge and in low-grade gliomas. By limiting the access to the tumor, therapies fail to achieve sufficient concentrations in vivo and to engage target limiting therapeutic effect [46]. There are some exceptions that can cross the BBB—notably temozolomide which is the current standard of care. Similar limitations also exist for immunotherapies and especially for large molecules like antibodies. The BBB becomes a challenge when the antibody is targeting something within the TME as opposed to systemic modulation [47,48]. The blood–tumor barrier (BTB) differs in its architecture and functionality relative to the normal BBB. Even though it is more permeable due to dysfunctional vessels, loss of pericyte uniformity of vessel coverage, and reduced cells tight junctions [49], it still limits drug delivery to the tumor. There is also functional and structural heterogeneity of the BBB/BTB within the same tumor and between different types of brain tumors (brain metastasis and gliomas) [50]. Thus, these factors should be taken into consideration for the development of future effective systemic therapies. In an animal model of gliomas, targeting pericytes and disrupting the BTP can enhance chemotherapeutic drug delivery to the brain tumor result in therapeutic efficacy [51].

To induce better drug delivery to the tumor, direct intracranial administration strategies have been conducted including clinical trials of convection enhanced delivery that have only shown modest prolongation of survival [52]. This approach is still being optimized and refined [53,54]. Intrathecal and intraventricular administration have also been conducted [55,56] but are still limited to disease in the leptomeningeal spaces or tumors residing in the ventricles [57,58] and do not necessarily benefit parenchymal disease. An emerging strategy for delivering immune therapeutics is BBB opening ultrasound. In preclinical studies, anti-PD-1 concentrations, and CAR T cell persistence in the TME were enhanced with associated increases in survival [42]. BBB opening ultrasound has already been evaluated in association with chemotherapeutic agents in several Phase I and II clinical trials, and has demonstrated promising efficacy and safety for use in patients with glioblastoma [59,60]. Whether BBB opening ultrasound can trigger immunological reactivity in the TME by itself is likely a function of factors such as the size and amounts of administered microbubbles and sonication parameters, among others, that will need to be optimized for a given strategy [61,62]. Several methods of optimization and imaging are ongoing for better control over the microbubble dynamics and optimal safety [63,64,65]. Tumor-specific factors such as the degree of enhancement, which is a function of vascular permeability, or the lack thereof, may also influence therapeutic delivery. Notably however are preclinical studies by Brighi et al. that have shown therapeutic delivery of antibodies to non-enhancing gliomas [66].

Immunotherapy molecular conjugates are also being developed as a way to enhance delivery across the BBB. The coupling of CTLA-4 and PD-1 to targeted nanoscale immunoconjugates on a natural biopolymer scaffold (poly β-L-malic acid) triggers prolonged survival in preclinical glioma models with better delivery of the drugs, and induced adaptive immune response including an increase of CD8^+^ T cells, NK cells and macrophages associated with a decrease in regulatory T cells (Tregs) in the brain TME [67]. Another aspect can be taken into consideration: cancer cells develop migratory functions, and express cell surface molecules that enable them to cross the BBB to colonize the brain parenchyma creating a brain metastasis. The study of such molecular components and functions of circulating cells permit the discovery of potential targets for therapies that will allow the disruption of the BBB to facilitate drug-delivery or by counteracting the metastatic effect of the cancer cells inhibiting them from breaching the BBB [68,69,70,71]. Different facets in the structure and functionality of the BTB/BBB are being studied to ameliorate access to brain parenchyma (Table 1).

## 6. Differential T Cell Deactivation and Suppression in CNS Cancer Lineages

Multiple immune suppressive mechanisms and pathways are utilized by the tumor cells and their environment to inhibit anti-tumor eradication. T cells are usually the effectors of the cytotoxic anti-tumor immune response in many cancers and if found in the glioblastoma TME, they are under a dysfunctional state and exhausted. Exhaustion occurs after repeated antigen exposure that leads to the expression of multiple immune checkpoints on the T cell surface [93]. Brain metastases are more enriched with TILs relative to glioblastoma that have abundant tumor-associated myeloid cells (TAMs) [8,27]. As such, engagement of immune checkpoints and modulation is more available for targeting in brain metastasis relative to glioblastoma. Notably, the expression of TIM-3 and LAG-3 immune checkpoints is relatively low in gliomas [11], and as such these therapeutic targets are unlikely to be beneficial for the treatment of patients with glioma for most patients and especially as a monotherapy. Both cytotoxic T-lymphocyte-associated protein 4 (CTLA-4) and programmed death 1 (PD-1), that are induced in activated T cells providing inhibitory signals by binding to their ligands expressed on the surface of antigen presenting cells or tumor cells (Figure 1), are more frequently expressed. Multiple clinical trials have shown that brain metastases are more responsive to immune checkpoint inhibitors therapies such anti-PD-1 and anti-CTLA-4 [6,94,95] relative to glioblastoma [96,97]. Other immune checkpoints, like TIGIT, were identified to be highly expressed on T cells, especially cytotoxic CD8^+^ tumor infiltrating T cells in murine models of glioma [19]. More recently, through an analysis of glioma infiltrating T cells, Mathewson et al. found that these cells express natural killer (NK) genes. They demonstrated an enhanced anti-tumor effect of these T cells through the direct blockade of CD161, a NK cell receptor expressed on the T cells surface, or by the inactivation of its respective NK gene KLRB1, highlighting the role of CD161 and potential therapeutic intervention [98]. The association of all these immunosuppressive markers on T cells surface induce a state of reduced responsiveness to immune stimulations and thus inhibited effector immune response.

A frequently expressed immune suppressive pathway in glioblastoma is the A2aR-adenosine pathway [99,100]. Related to this pathway is CD73, an enzyme shown to be preferentially expressed by tumor cells in glioma patients [10] and upregulated in immune cells [11], especially in IDH1-mutant gliomas. This ectonucleotidase is responsible for the activation of adenosine that binds to the A2a receptor on both T cells and myeloid cells thereby triggering immunosuppression [99] (Figure 1). Therefore, this pathway is a potential therapeutic target. However, only limited therapeutic benefits were shown by administration of adenosine receptor inhibitors in multiple murine models of gliomas, including those expressing CD73 even in combination with other inhibitors [11]. This highlights the challenges in approaches aimed to reverse T cell exhaustion induced in glioma TME [19,38].

## 7. Emerging Strategies for Modulation of Immunosuppressive Tumor Associated Macrophages

TAMs, the most abundant immune cells present in the TME, express a wide variety of immunosuppressive phenotypes [101]. Gliomas express CD47 which blocks phagocytosis thereby evading immune recognition and eradication [102,103]. CD47 blockade in several murine models [102,104] has shown therapeutic benefit by freeing the innate immune system to activate APCs thereby leading to effector CD8^+^ effector T cells mediated immune responses [105,106]. However, monotherapy leads to treatment resistance [106,107,108]. Therefore, combination therapies can represent a solution for better induced anti-tumor responses and outcomes. A recent study demonstrated that combining anti-CD47 targeted therapy with temozolomide creates a pro-phagocytosis effect and induces antigen presentation by activating and upregulating the cGAS-STING pathway in APCs, thus leading to an effective adaptive immune response [108].

Stimulator of interferon genes (STING) is an important component of the innate immune response to pathogenic DNA. It is a widely expressed sensor of cellular stress, activated by cytosolic cyclic dinucleotides, which may be released by bacteria or created through cytosolic self- or viral-DNA interaction with cyclic GMP-AMP synthase (cGAS) [109,110]. STING agonists can induce T cell infiltration into tumors known to be devoid of such immune composition and in tumors enriched with myeloid immune cells through pro-inflammatory activation resulting in marked in vivo therapeutic activity [111,112]. This pathway bridges the innate and adaptive immune systems both by triggering interferon (IFN) release and through activation of myeloid cells (Figure 1). Distinct from most other innate immune agonists, STING activation can re-educate tumor-supportive immunosuppressive macrophages toward a pro-inflammatory phenotype and can reverse the suppressive phenotype of myeloid-derived suppressor cells (MDSCs) [113,114]. STING agonists have demonstrated radiographic responses in canines with high-grade gliomas [115]. In summary, STING agonists may be a compelling therapeutic strategy for gliomas because they: (1) can simulate a foreign body reaction, thus providing a “target”; (2) induce IFN, thereby providing potent T cell effector action; (3) induce chemokine production and thus T cell trafficking to the tumor; and they are (4) easy to synthesize.

The reciprocal immune modulatory strategy of TAMs is to block a key deactivating pathway the signal transducer and activator of transcription 3 (STAT3). STAT3 is an important mechanism of suppression of both innate and adaptive components of the immune system (Figure 2). STAT3 expression shifts the TAMs to an immunosuppressive phenotype secreting immunomodulatory suppressive factors like IL-10 and TGF-β, and impairing phagocytosis and antigen presentation [116,117,118]. Similarly, STAT3 impairs maturation and antigen presentation by dendritic cells (DCs) preventing T cell activation and proliferation [119,120]. The inhibition of STAT3 can reactivate the immune system in the TME by promoting infiltrating DCs maturation, increasing expression of the co-stimulatory molecules (CD80/CD86) necessary for T cell activation, and decreasing the number of myeloid derived suppressor cells (MDSCs) in the immune microenvironment [121,122,123]. Additionally, STAT3 is a key inducer of immune suppressive cytokines (IL-10, IL-4, IL-6 and TGF-β), maintains immunosuppressive cell cross-talk [124,125,126,127,128], increases tumor infiltration by MDSCs, and induces T cell arrest and apoptosis [129,130,131,132]. Through MDSCs secretion of INF-α and other mechanisms, STAT3 upregulates the expression of inhibitory immune checkpoints like PD-L1 on the surface of TAMs and tumor-infiltrating DCs [126,130]. Furthermore, STAT3 signaling is correlated with a decrease in effector T cells infiltration of the tumor and prevents CD8^+^ T cell activation by increasing the secretion of INF-γ [133]. STAT3 activation correlates with the activation and upregulation of FOXP3 in T cells and is a key inducer of immunosuppressive Tregs infiltration of the TME [134,135,136,137]. Several studies, in mice models of glioma and on patient-extracted glioma tumor cells, have demonstrated the benefits of inhibiting STAT3 that enhances the anti-tumor immune response by improving T cell, DCs and NK activation in the TME [119,138].

The combination of the standard of care radiation therapy with a STAT3 inhibitor, showed improvement in the overall survival of mice harboring intracranial gliomas [139]. This was shown to be immunologically mediated with reprogramming of the DCs in the TME. STAT3 signaling is involved in a variety of intrinsic and acquired resistance mechanisms to administered anti-tumor therapies such as temozolomide [140,141,142], radiation [143,144,145,146,147,148,149], and targeted therapies [150,151,152]. As such, some studies demonstrated that the combination of anti-VEGF (Cediranib) and STAT3 inhibitors (AZD1480) for the treatment of glioblastoma in a mice model led to a decrease in the tumor volume and angiogenesis [153,154]. The co-inhibition of STAT3 and MET induce glioma tumor cells destruction by reactivating apoptosis mechanisms [155]. As such, an area of future investigation is combinatorial immune modulatory or chemotherapeutic strategies with STAT3 inhibition.

## 8. Conclusions

Therapy with a single modality is unlikely to achive meaningful improvements in the outcomes of patients with glioblastoma. In order to maximize the effect of current immunotherapies, several complimentary potential solutions will need to be deployed such as cytoreductive surgery and other measures to reduce immune suppression [156]. Immune activation [157,158,159], in combination with other therapies like radiation [160] may facilitate and trigger extracranial T cells priming for a better anti-tumor immune response. Future combination therapies using different treatments including facilitated delivery, may offer a solution for addressing complexity and heterogeneity of the TME in glioblastoma.

## Figures and Tables

**Figure 1 cells-10-02032-f001:**
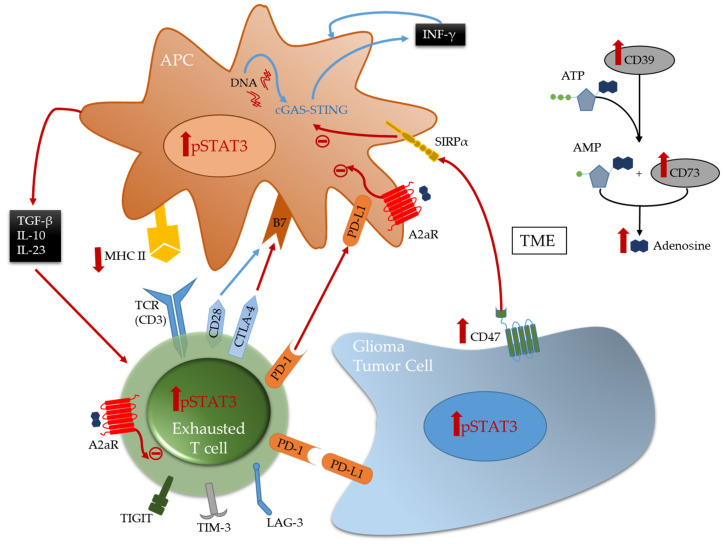
Immunosuppressive cross-talk between the immune system and glioma tumor cells. This figure summarizes the different immune checkpoints and cellular pathways associated with T cell exhaustion and inducing the immunosuppressive phenotype of antigen presenting cells (APC). Expression of CTLA-4 immune checkpoint on the T cell surface and its binding to B7 (CD80/CD86) on the APC surface, competes with CD28 thereby blocking T cell activation. In addition, the expression and upregulation of various immune checkpoints such as PD-1, TIM-3, and LAG-3 on the T cell surface triggered by repeated exposures to tumor antigens induces T cell exhaustion. Upregulation of immune checkpoints on immune and glioma cells can be induced by the upregulation of nuclear p-STAT3 known to be a molecular hub for the induction of immunosuppression. pSTAT3 downregulates the major histocompatibility II (MHC II) on the APC surface and induces the secretion of immunosuppressive cytokines like TGF-β, IL-10, and IL-23 (known to shift T cells differentiation to T-regs) that inhibit T cell activation and create an immunosuppressive environment. Glioma cells can express CD47 that binds to the SIRPα receptor on the APC surface thereby impeding phagocytosis. The cGAS-STING pathway senses DNA in the cytoplasm thereby stimulating the immune system by secretion of pro-inflammatory INF-γ favorizing immune cells activation and a macrophage pro-inflammatory phenotype shift. The presence of both CD39 and CD73 in the TME triggers the production of adenosine which binds to the A2aR receptor expressed on the surface of the immune cells thereby inducing immunosuppressive pathways rendering the immune system tumor protective.

**Figure 2 cells-10-02032-f002:**
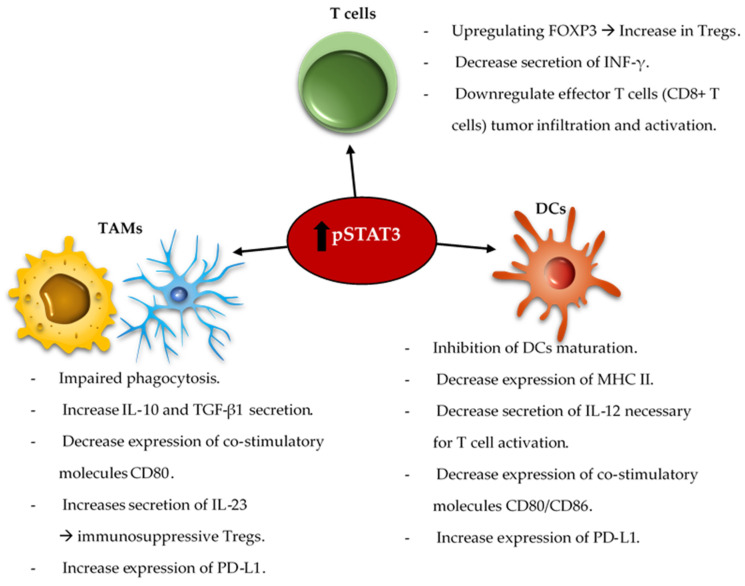
Summary of the immunosuppressive role of pSTAT3 in the innate and adaptive immune systems. The innate immune system includes tumor-associated macrophages (TAMs) and dendritic cells (DCs); whereas the adaptive immune system includes T cells. The upregulation of pSTAT3 in the cell nucleus due to various stimuli leads to activation and inhibition of different pathways. In TAMs, increase in pSTAT3 induces impairment of phagocytosis, increases secretion of immunosuppressive cytokines like IL-10, TGFβ-1, and IL-23, and it also induces the expression of PD-L1 on TAMs and DCs. In DCs, pSTAT3 inhibits maturation and leads to a decrease in major histocompatibility II (MHC-II) and co-stimulatory molecules CD80/CD86 expression on the cell surface. In addition, it leads to a decrease in DCs secretion of IL-12 necessary for T cell activation. Finally, the expression of pSTAT3 in T cells leads to effector response inhibition and induction of Tregs by upregulating FOXP3 nuclear expression, thereby leading to a decrease in the secretion of INF-γ and other pro-inflammatory cytokines.

**Table 1 cells-10-02032-t001:** Summary of non-invasive or minimally invasive to invasive approaches used to improve drug delivery across the BBB/BTB to brain parenchyma.

**Non-to minimally invasive approaches**	**Physical**		Focused ultrasound (FUS) of the region of interest with microbubble IV injection [42,59,71,72,73].
			Radiation of the region of interest [74,75]. Radiation therapy in combination with FUS [76,77].
	**Biological**	Viral	Viral Vaccines: - Adenovirus mediated gene therapy [78]. - Polio virotherapy [34]. - Live attenuated Zika virus vaccine (ZIKV-LAV) [79].
		Cellular	- Immune cells: E-selectins, P-selectins, CAMS, integrins, chemokines and cytokines. - Tumor cells: L1-CAM, COX2, HBEGF, MMP9, VEGF, cathepsin S [80] and 2,6-sialyltranferase ST6GALNAC5 on breast cancer metastatic cells [81]. - Stem cells: CD44, CD99, integrins, CXCR4, CCR2 and VCAMs.
		Molecular	- Receptor-mediated drug transport modulation: transferrin, insulin, and insulin-like growth factor 1 receptors [47,82]; Lipoprotein receptor-related protein 1 (LRP1) targeted peptide for chemotherapy delivery [83,84,85]. - Carrier-mediated transcytosis: targeting glucose transporter GLUT1 and LAT1 on endothelial cells [86]. - Targeting efflux pump modulation: ABC transporters inhibitors (P-gp and BCRP inhibitors for better delivery of chemotherapy and various targeted therapies) [87,88,89,90]. - Protein Kinase Inhibitors (PI3K and mTOR inhibitors that reduce liability of active efflux by P-gp and BCRP and favorise passive diffusion of drugs) [88,91,92]. - Tumor-targeted polymer-conjugated checkpoint inhibitors: CTLA-4 and PD-1 attached to nanoscale immunoconjugates on natural biopolymer scaffold (poly β-L-malic acid) [67].
**Invasive approaches**			Direct intracranial injections to the tumor parenchyma via a cannula or CSF-tumor cavity reservoirs [53,54].
			Intra-ventricular and intrathecal injections to the leptomeningeal space [57,58].

## Data Availability

Not applicable.

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
