# Peer review of "Immune Microenvironment Landscape in CNS Tumors and Role in Responses to Immunotherapy"

_cells, 2021, doi:10.3390/cells10082032_

Round 1

Reviewer 1 Report

The review by Najem et al. discusses current immunotherapeutic concepts and their underlying molecular rationales for the treatment of CNS tumors. The review is very comprehensive and well-written. I have only the following minor points:

  • Please correct the following typo: … is ‘likely due to’ their ‘sequestration’ in the bone marrow [18], exhaustion, and … (line 64)
  • ‘… On the other hand, IDH-mutant tumors are enriched with tumor-resident microglia, which is in contrast to IDH wild-type and brain metastasis that are infiltrated with monocyte-derived macrophages originating from the periphery [8, 24] (line 98). I would suggest to also cite Friedrich, Nat Cancer, 2021
  • ´… Gliomas, especially glioblastomas, are typically described as having a “cold” micro-environment given the paucity of TILs displaying an effector response. Molecularly distinct gliomas, depending on their IDH mutation status, have different immune compositions and landscape that defines its TME' (line 90). I would suggest to also cite Kohanbash, JCI, 2017 and Bunse, Nat Med, 2018, Friedrich, Nat Cancer, 2021
  • ‘… Additional evidence that the glioma TME is distinct from other cancer lineages is the lack of tumor mutational burden (TMB) being a predictive response biomarker to immune checkpoint inhibitors. McGrail et al. have recently shown in an analysis of over 10,000 patients that not all tumors with high-TMB developed effector immune responses to immune checkpoint inhibitors therapy.' (line 123). I would suggest to also cite Touat, Nature, 2020
  • ‘… Other immune checkpoints, like TIGIT, were identified to be highly expressed on T Cells, especially cytotoxic CD8+ tumor infiltrating T cells in murine models of glioma [19] (line 253). I would suggest to also briefly discuss and cite Mathewson, Cell, 2021
  • ‘…reactivity in the TME in of (rephrase 'in of' to 'by') itself is likely a function of factors such as the size and amounts ('of') administered microbubbles and sonication parameters, among others, will need to be optimized for a given strategy [57, 58] (line 214). Likely, also the tumor itself is a factor such as enhancing vs. non-enhancing lesions/tumors which could be briefly discussed.

Reviewer 2 Report

The paper by Najem and colleagues is a very well-written, detailed and interesting review article. The authors review up-to-date aspects of the immune response to gliomas, namely the location of immune cells in the tumor microenvironment and the key mechanisms that modulate the anti-tumor immune response. They also discuss how these aspects of the immune response impact immunotherapy.  The figures are clear and informative. Overall, an interesting read and of great interest to the field. 

Author Response

Thank you.

Reviewer 3 Report

The authors have provided a very well-organized, well-written, comprehensive review of the immune microenvironment in gliomas, and the current limitations of immunotherapy in gliomas. Indeed, immune checkpoint inhibition as monotherapy has not been effective in the treatment of this disease. Combination strategies have certainly been highlighted here. 

Author Response

Thank you.